# Translesion synthesis polymerases are dispensable for *C. elegans* reproduction but suppress genome scarring by polymerase theta-mediated end joining

Ivo van Bostelen[1], Robin van Schendel[1], Ron Romeijn[1], Marcel Tijsterman[1,2]*

**1** Department of Human Genetics, Leiden University Medical Center, Leiden, The Netherlands, **2** Institute of Biology Leiden, Leiden University, Leiden, The Netherlands

* m.tijsterman@lumc.nl

**Data Availability Statement:** Sequence data can be found in the NCBI SRA repository under project number: PRJNA605924. Some sequences have

## Abstract

Bases within DNA are frequently damaged, producing obstacles to efficient and accurate DNA replication by replicative polymerases. Translesion synthesis (TLS) polymerases, via their ability to catalyze nucleotide additions to growing DNA chains across DNA lesions, promote replication of damaged DNA, thus preventing checkpoint activation, genome instability and cell death. In this study, we used *C. elegans* to determine the contribution of TLS activity on long-term stability of an animal genome. We monitored and compared the types of mutations that accumulate in REV1, REV3, POLH1 and POLK deficient animals that were grown under unchallenged conditions. We also addressed redundancies in TLS activity by combining all deficiencies. Remarkably, animals that are deficient for all Y-family polymerases as well as animals that have lost all TLS activity are viable and produce progeny, demonstrating that TLS is not essential for animal life. Whole genome sequencing analyses, however, reveal that TLS is needed to prevent genomic scars from accumulating. These scars, which are the product of polymerase theta-mediated end joining (TMEJ), are found overrepresented at guanine bases, consistent with TLS suppressing DNA double-strand breaks (DSBs) from occurring at replication-blocking guanine adducts. We found that in *C. elegans*, TLS across spontaneous damage is predominantly error free and anti-clastogenic, and thus ensures preservation of genetic information.

## Author summary

Research in the fields of DNA repair and mutagenesis has led to enormous insight into the mechanisms responsible for maintaining genetic integrity. However, which processes drive *de novo* mutations and will thus contribute to inherited diseases are still unclear. One process thought to underlie spontaneous mutagenesis is replication of damaged DNA by specialised so-called "Translesion synthesis" polymerases, which have the ability to replicate across damaged bases, but are not very accurate. To address the impact of TLS or the lack thereof on genome integrity, we have knocked out all TLS enzymes that are

been previously published and can be found in the NCBI SRA repository under project numbers: PRJNA196232 and PRJNA260487.

**Funding:** RvS and MT are supported by grants (ALWOP.269 and ALWOP.393) from the Netherlands Organisation for Scientific Research (NWO, https://www.nwo.nl/en). The funders had no role in study design, data collection and analysis, decision to publish, or preparation of the manuscript.

**Competing interests:** The authors have declared that no competing interests exist.

encoded by the *C. elegans* genome, individually and in combination, and monitored mutation accumulation during prolonged culturing of these animals without external sources of DNA damage. We found that TLS is not the major driver of spontaneous mutagenesis in this organism, however, it protects the genome from harmful small deletions that result from mutagenic repair of DNA breaks. We also found that, contrary to what was expected, TLS activity is not essential for reproduction in a multicellular organism with the tissue complexity and genome size of *C. elegans*.

## Introduction

Although mutagenesis is a prerequisite for evolution, mutations are also life threatening as they are at the basis of inborn diseases and age-related pathologies such as cancer. To sustain genetic integrity several mechanisms have evolved that restrict the level of mutation induction. For example, during DNA replication, the combined action of proofreading activity of the replicative polymerases and the mismatch repair pathway provides an estimated 10.000-fold increase in copying accuracy [1,2]. Apart from errors during replication of non-damaged DNA, mutations can also result from incorporation of nucleotides across damaged bases that are caused by exogenous sources or endogenous processes within the cell. For example, oxidative metabolites can react with DNA, which then hampers DNA replication [3]. Efficient and unperturbed DNA synthesis is nevertheless essential for maintaining genetic integrity because persistent replication impairment can lead to under-replicated areas of the genome, and the formation of highly toxic DNA double-strand breaks (DSBs) that may result in genomic rearrangements or cell death. Such detrimental outcomes are profoundly suppressed by the action of the excision repair pathways base excision repair (BER) and nucleotide excision repair (NER), which can replace the damaged bases, thereby preventing these from becoming replication obstacles [3,4]. DNA damage can, however, also be bypassed through so-called DNA damage tolerance pathways, in which lesions are left unrepaired, yet replication of the genome is completed, hence ensuring cell cycle progression [5]. A well-studied mechanism to tolerate DNA damage is translesion synthesis (TLS). Despite the potential of replicative polymerases to bypass certain DNA lesions (*e.g.* [6]) they are blocked by a plethora of damaged bases, and specialized TLS polymerases are required to synthesize DNA opposite these impediments. Lesions can be bypassed directly when the replicative polymerase is temporarily exchanged for a TLS polymerase at the replication fork during S-phase, or, alternatively, single-strand DNA gaps temporarily remain at the site of base damage and bypass and gap filling occurs after S-phase completion [7, and references therein].

In eukaryotes TLS is mediated by Y-family polymerases Polη, Polκ, Polι and Rev1 and the B-family polymerase Polζ, the latter containing a catalytic subunit, Rev3, a regulatory subunit, Rev7 [8,9], as well as accessory subunits [10–12]. These TLS polymerases lack proofreading activity and have wide catalytic centers to allow for replication across damaged bases and DNA synthesis from misaligned primer termini. These characteristics cause TLS polymerases to have lower fidelity than replicative polymerases, making them inherently error prone. Whereas some types of lesions require only a specific Y-family polymerase, other types require the sequential action of two or more TLS polymerases [13–15].

While the molecular details of TLS have become better understood it remains unclear how TLS action affects genome maintenance or influences spontaneous mutagenesis either positively or negatively, on a genome-wide scale and in time. The model system *C. elegans* is well suited to address these questions for multicellular organisms, because of a condensed genome

and the ability for clonal propagation, which makes whole genome sequencing (WGS) practical. In previous studies we have described how *C. elegans* Y-family polymerases Polη and Polκ contribute to genomic stability [16–18]. Here, we investigate the contribution of REV1 and REV3 in suppressing genome alterations, and we address the redundancy between the different TLS enzymes by monitoring mutation accumulation in animals that lack either all *C. elegans* Y-family polymerases or all TLS activity. Our study presents the most comprehensive analysis of how TLS activity affects the stability of a genome under non-challenged circumstances.

## Results

### Generation and characterization of *rev-1* alleles

To study the role of REV-1 in the maintenance of genomic stability we analyzed several mutant alleles: *rev-1(gk455794)* was generated by the million mutation project [19] and has a point mutation in the acceptor splice site of exon 7, and two early stop alleles were obtained through targeting exon 2 via CRISPR/Cas-9 technology (S1 Fig). We also isolated an allele (named *rev-1BRCT*) containing a G283>D amino acid substitution in the evolutionarily conserved BRCT domain. G283 of *C. elegans* REV-1 aligns to G193 of yeast REV1 and G76 of mice Rev1, which in those species are essential for the functionality of the BRCT domain [20–22].

All *rev-1* mutant strains had brood sizes and embryonic survival frequencies comparable to wild-type controls (Fig 1A and Fig 1B), demonstrating that REV-1 is not essential for animal development and fertility under unchallenged conditions. However, similar to REV1 deficient yeast and mES cells [23–25], REV-1 deficiency in *C. elegans* confers hypersensitivity to UV-induced DNA lesions, which manifests as reduced embryonic survival (as compared to wild-type controls) when hermaphrodites are exposed to UV-C (Fig 1C). While a similar degree of hypersensitivity is noticed for the three putative knockout alleles, the *rev-1BRCT* allele confers an intermediate phenotype consistent with a partial loss of functionality. This evolutionary conserved UV-hypersensitivity of REV1 deficiency is less dramatic than observed for *C. elegans* mutants defective in Polη, another TLS polymerase implicated in the bypass of UV-induced damage [16,26].

### REV-1 suppresses the formation of genomic deletions and rearrangements

When TLS is impaired, replication forks can collapse at sites of base damage and form DSBs. In the mitotic compartment of the *C. elegans* gonad, recombinogenic DNA substrates can be visualized by staining for the recombinase RAD-51, which forms foci. While such foci are not found in wild-type animals, both *rev-1(gk455794)* as well as *rev-1BRCT* animals had a very small, yet statistically significant number of nuclei in the mitotic zone containing a RAD-51 focus (Fig 1D and Fig 1E). To address potential mutagenic consequences of elevated levels of replication stress or DNA breaks, as suggested by increased RAD-51 foci formation, we assayed mutation induction in *rev-1* mutants. We first used the phenotype-based *unc-93* reversion assay [27,28]: animals carrying the toxic *unc-93(e1500)* allele have a very poor capacity to move and also grow slower. However, any additional mutation in *unc-93* leading to complete loss of function will lead to reversion of these phenotypes. Loss of function mutations in the suppressor genes *sup-9*, *sup-10*, *sup-11* or *sup-18* will also revert *unc-93(e1500)* animals to wild-type-resembling growth and movement. Picking wild-type-moving worms out of populations of *unc-93* animals, followed by inspection of their genomes thus results in a mutation profile. To establish such profiles, we isolated 30 revertants for each genotype and subsequently sequenced the *unc-93* locus. In WT, *rev-1BRCT* and *rev-1(gk455794)* animals we respectively found 7, 14, and 14 causative mutations in *unc-93*(*e1500*) (Fig 1F); the remaining revertant

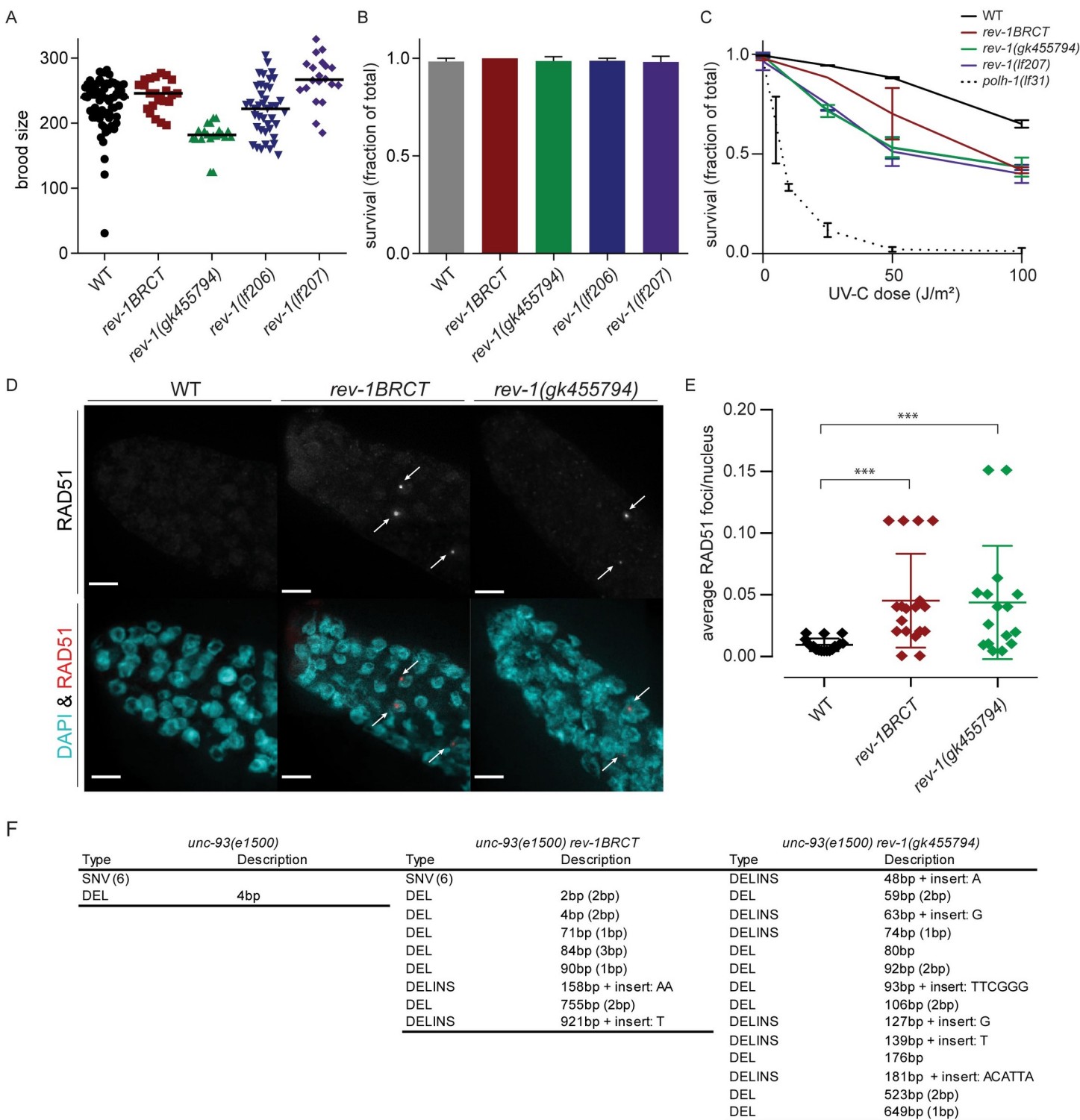

**Fig 1. Spontaneous DNA damage and mutagenesis in *rev-1* deficient animals.** A) Quantification of total brood sizes for different *rev-1* alleles. Each data point represents the total brood of a single hermaphrodite. B) The surviving fraction of progeny was calculated as the number of hatched eggs divided by the brood size. Error bars denote SD. C) Progeny survival in response to UV-C exposure to hermaphrodites of the indicated genotype. Error bars denote SEM. D) Representative images of RAD-51 and DAPI-stained distal tips of *C. elegans* gonads. Arrows point to RAD-51 foci. E) Quantification of spontaneous RAD-51 foci in the mitotic compartment of the gonadal arms. Each dot represents the average number for a single mitotic region. Error bars indicate mean with sd (*** indicates p<0.001, t-test). F) Molecular descriptions of the mutations found in *unc-93* for the indicated genotype (SNV = single-nucleotide variation, DEL = deletion, DELINS = deletion with insert). The number of SNVs found is indicated between parentheses. The degree of micro-homology found at deletion junctions is indicated between parentheses. For *rev-1* deficient animals 30 revertants were analyzed. Data from REV-1 proficient *unc-93* reversion is retrieved from [13].

animals likely carry mutations in the suppressor loci, which were not further analyzed. In the WT background, 6 out of 7 mutations were single nucleotide variations (SNVs) that disrupt gene function via amino acid substitution, introduction of an early stop or loss of a splice site; one deletion of 4 bp was found. Similar base substitutions and deletions of few bases were also found in the *rev-1BRCT* mutant, but here, 6 of the 14 mutations were larger deletions (>50 bp). For *rev-1(gk455794)*, all causative mutations (n = 14) were deletions >50 bp in size. From this data we conclude that REV-1 has an important role in suppressing a specific type of spontaneous mutagenesis, *i.e.* the formation of small genomic deletions.

## Mutagenic tradeoff by REV-1 action

While the *unc-93* reversion assay is an established method to study mutagenesis in *C. elegans* it has drawbacks. For instance, the mutational target is limited and fixed, and the only mutations that will be picked up are those that disrupt gene function, thus creating a bias: deletions are more likely to disrupt gene function than SNVs. Also, the assay conditions themselves hinder accurate quantifications of mutation rates. To overcome these shortcomings and study spontaneous mutagenesis in a predominantly unbiased and quantitative way, we performed whole-genome sequencing (WGS) of wild-type and *rev-1* mutant animals that were clonally grown for multiple generations (Fig 2A and S1 Table; S2 Table). For wild-type animals we found mutation rates to be ~0.23 SNVs, ~0.04 micro-satellite indels, and ~0.04 small deletions per animal generation. While the rate for micro-satellite indels is identical in *rev-1* mutant animals, the SNV rate is lowered; the SNV spectrum, however, was similar, if not identical, to that in wild-type animals (Fig 2B). Interestingly, the reduction in SNVs matched an increase in the rate of deletion formation in the >50 bp size range (Fig 2A and Fig 2C): we identified 15 cases in *rev-1(gk455794)* genomes and 1 in the wild-type control, constituting a changed distribution (p = 0.0032, Mann-Whitney). This outcome leads to the suggestion that base damage requiring REV-1 action is mutagenic both in the presence and absence of REV-1: when bypassed by REV-1-mediated TLS a SNV results, but in REV-1 compromised cells a deletion is the outcome. In addition to simple deletions a limited number of complex rearrangements was found in *rev-1* mutant animals–these were not observed in the analyzed genomes of wild-type animals.

This mutation profile of *rev-1* animals strikingly resembles the ones we previously described for strains deficient in TLS polymerases POLH-1 and POLK-1, as well as for strains defective for the helicase FANCJ/DOG-1 [17,29,30]. Deletion mutagenesis in *polh-1* and *polk-1* strains occur seemingly random throughout the genome, whereas deletions in *dog-1* animals map to sequences that are able to fold into replication-blocking G-quadruplex structures. Deletions in *rev-1* mutant genomes do not map to particular sequence motifs, despite potential indications from genetic analysis in chicken DT40 cells, which demonstrated a function for REV1 in replicating through G-quadruplex structures [31,32]. We also tested potential synergy between *rev-1* and *dog-1* with respect to G-quadruplex stability, but did not find any (S2 Fig).

## TMEJ repairs DSBs that result from REV-1 loss

The similarity in the profile of spontaneous mutagenesis in *rev-1* and *polh-1* or *dog-1* points to a causal involvement of polymerase theta-mediated end joining (TMEJ). TMEJ was previously shown in *C. elegans* to repair DSBs that arise at replication-impeding base damages or secondary structures [17,30]. Indeed, deletions accumulating in *rev-1* have signature hallmarks of TMEJ: micro-homology at break sites and occasionally inserts that are homologous to sequences in the immediate vicinity of the deletion [17, 30, 33, 34]. To address potential causality with respect to TMEJ involvement we used the *unc-93* reversion assay and determined the

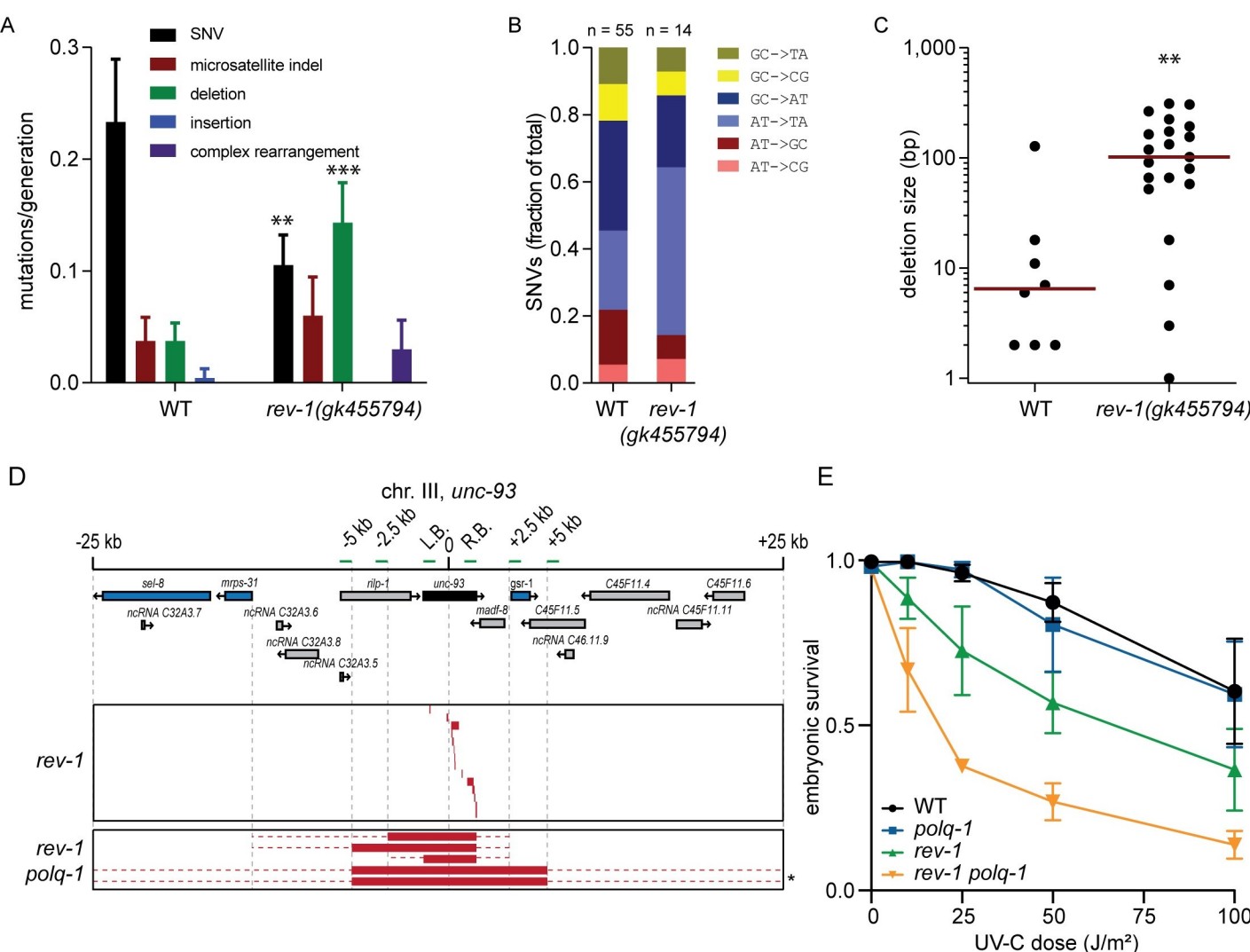

**Fig 2. REV-1 suppresses genomic scars caused by TMEJ, without compensation by other Y-family TLS polymerases.** A) WGS analysis of wild-type (WT) and *rev-1* deficient animals that were grown for 40–60 generations under unperturbed conditions. Per genotype at least three independent populations were analyzed. Mutations were categorized into five different classes (** indicates p<0.01, *** indicates p<0.001, t-test). B) distribution of SNV types. C) Deletions found in WGS data plotted by size (log10 scale) for both wild-type and *rev-1* deficient animals (** indicates p<0.01, Mann-Whitney). D) A schematic representation of the genome surrounding the *unc-93* locus on chromosome III. Essential genes are colored blue, the *unc-93* gene is colored black, while other genes are in grey. PCR amplicons used to estimate deletions sizes are indicated in green. Underneath, boxed in red a visual representation of *unc-93* loss of function alleles derived from the indicated genotype. All *unc-93* alleles obtained from *rev-1* animals were determined at nucleotide resolution via Sanger sequencing. For the sizable *unc-93* alleles that accumulated in *rev-1 polq-1* the minimal (red box) and maximal (red dotted line) borders were established. * denotes animals that did not produce viable progeny, likely due to a homozygous mutation in the essential gene *gsr-1*. E) Progeny survival in response to UV-C exposure to hermaphrodites of the indicated genotype. Error bars denote SEM.

molecular nature of the mutations that spontaneously arise in animals mutated for both *rev-1* and *polq-1* (encoding pol theta/Polθ). We indeed found a dramatically changed mutation profile (Fig 2D). Instead of the characteristic 50-500bp size, all deletions isolated from *rev-1 polq-1* populations were profoundly larger: none that mapped to the *unc-93* locus retained the 2.5 kb spanning *unc-93* gene; 2 of 5 being >10 kb and having lost all loci between *unc-93* and the most nearby essential genes. To further asses the notion that TMEJ acts to repair failed TLS of damaged bases, we tested Polθ's involvement in relation to UV-induced DNA lesions. While Polθ deficiency itself does not sensitize otherwise wild-type animals to UV-C, it exacerbates the hypersensitivity of *rev-1* animals (Fig 2E).

We conclude that *C. elegans* REV-1 prevents DSB-induced deletion mutagenesis during unperturbed growth.

## REV-1 independent functions of POLH-1 and POLK-1

We previously analyzed knockout alleles of *polh-1* and *polk-1*, the genes encoding the other two Y-family polymerases present in the *C. elegans* genome: Polη and Polκ [16, 17]. While *polh-1* mutants, like *rev-1*, display some spontaneous mutagenesis under non-perturbed growth, *polk-1* mutant animals do not. However, the analysis of animals deficient for both genes revealed profound functional redundancy for these proteins: the marginal deletion mutagenesis observed in *polh-1* mutant animals dramatically increased when also POLK-1 was lost. To investigate whether one of these two TLS polymerases, or both, also provide back-up functions in REV-1 deficient animals (or *vice versa*) we generated *polh-1 polk-1 rev-1* triple mutant animals, which thus lack all Y-family TLS activity (the *C. elegans* genome does not encode Polι). Under unchallenged growth conditions, we found brood size and embryonic survival of these so-called "Y-family polymerase dead" mutants to be lower than wild-type, yet remarkably mildly affected (Fig 3A and Fig 3B). Because of this only mildly disturbed animal fitness we could grow animals for 50 generations to address mutation accumulation, and found that the mutation load in *polh-1 polk-1 rev*-1 mutants is very similar if not identical to that in *polh-1 polk-1* mutants (Fig 3D and S1 Table; S2 Table). For comparison we also included our previously published data for *polk-1* and *polh-1* mutant animals, which we re-analyzed using updated software.

From the observations that REV-1 deficiency brings about only a marginal level of deletion mutagenesis, and does not further enhance the genomic instability in *polh-1 polk-1* animals, we conclude that Polη and Polκ redundantly act to bypass spontaneous DNA damage in *C. elegans* in a *grosso modo* error-free manner. In such manner, these polymerases do not require the action of REV-1, yet are potentially recruited to sites of DNA damage by PCNA ubiquitylation [35].

## REV-1 suppresses genomic deletions through polymerase ζ

In addition to direct bypass through catalysis [36–39], REV1 also plays a non-catalytic role via interactions with other TLS proteins such as Polη and the REV7 subunit of B-family TLS polymerase Polζ [40–42]. Rev1 and Rev3 mutant cells have similar phenotypes, arguing that Rev1 is especially important for Polζ activity. To address whether the increased deletion mutagenesis in *C. elegans rev-1* mutants is a consequence of Polζ disfunction, we analyzed *rev-3* mutant animals. A putative null allele, *rev-3(gk919715)*, generated by the million mutation project, contains a nonsense mutation in exon 3 producing an early stop N-terminal of all relevant functional domains. (S1 Fig). Subsequent to extensive backcrosses in order to remove background mutations, we measured a reduced brood size, but no embryonic lethality for *rev-3 (gk919715)* (Fig 3A and Fig 3B). To validate a TLS deficiency in this genetic background, we exposed hermaphrodites to UV-C, and found hypersensitivity of progeny embryos comparable to the hypersensitivity observed for *rev-1* mutants (Fig 3C). Animals deficient for both REV-1 and REV-3 are as sensitive as either single mutant indicating no role for REV-1 outside Pol ζ in bypass of UV-induced damage. We next grew multiple *rev-3* mutant lines clonally for 50 generations after which their genomes were sequenced (S1 Table; S2 Table). We found that SNVs and indels at microsatellite repeats accumulate at the same rate in REV-3 proficient and deficient animals (Fig 3D), arguing that REV-3-dependent TLS on spontaneous lesions does not cause substantial numbers of point mutations or that backup mechanisms are equally or more mutagenic. However, and similar to the genomes of clonally propagated *rev-1* mutant

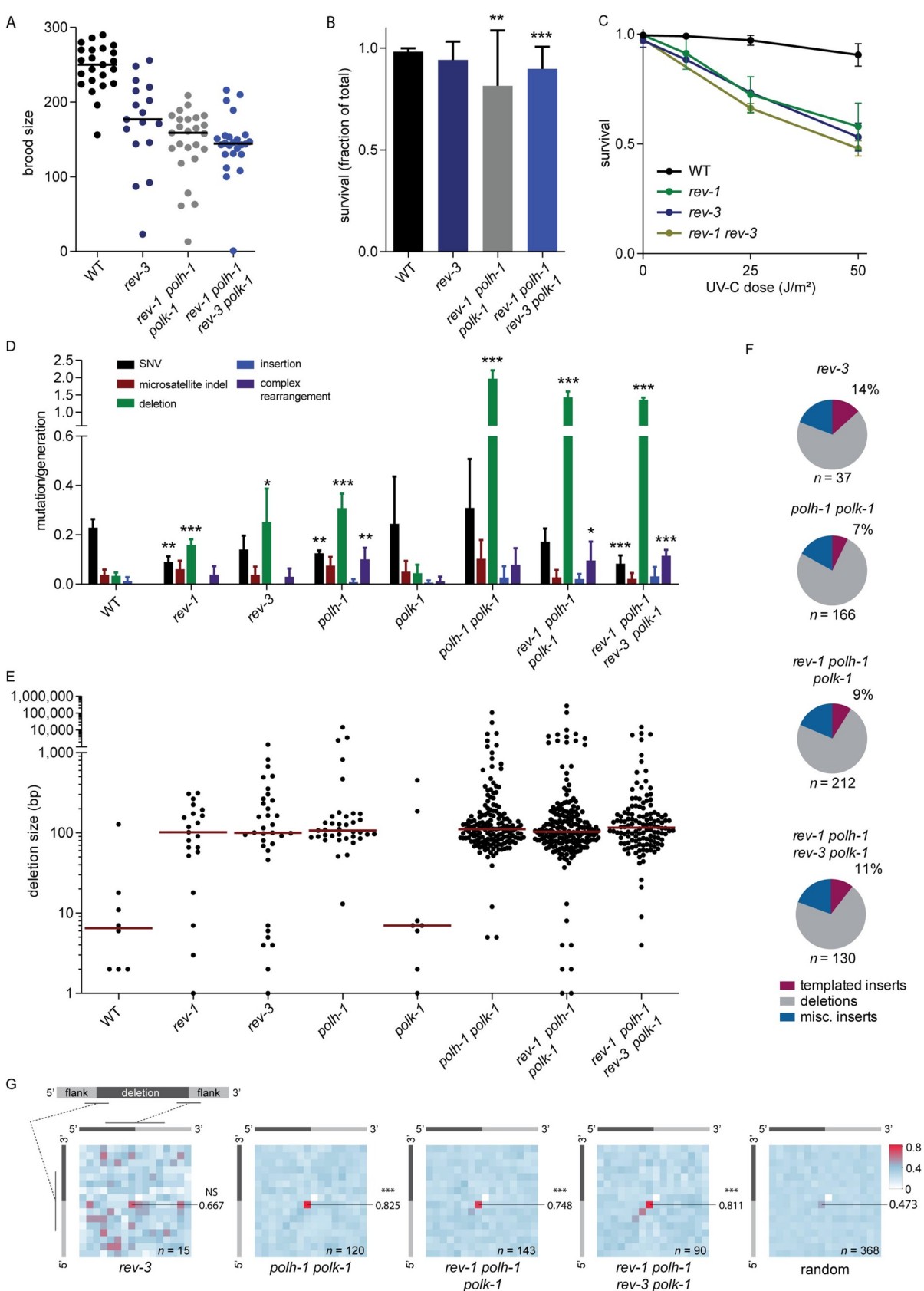

**Fig 3. Genome protection by REV-3 and the Y-family of TLS polymerases.** A) Quantification of total brood size for wild-type, *rev-3*, *rev-1 polh-1 polk-1* and *rev-1 polh-1 polk-1 rev-3* deficient animals. Each data point represents the total brood of a single hermaphrodite. B) The surviving fraction of progeny was calculated as the number of hatched eggs divided by the brood size. Error bars denote SD (** indicates p<0.01, *** indicates p<0.001, t-test). C) Progeny survival in response to UV-C exposure to hermaphrodites of the indicated genotype. Error bars denote SEM. D) WGS analysis of WT and TLS deficient animals grown for 20–60 generations. Per genotype at least three independent populations were analyzed, except for *polh-1* where only two independent populations were analyzed. Error bars denote SD (* indicates p<0.05, ** indicates p<0.01, *** indicates p<0.001, t-test). E) Size plot for deletions that accumulate in the genomes of the indicated genotypes. Red line represents median size (log10 scale). F) Pie charts visualizing the subdivision of all deletions into three categories: i) simple deletions, ii) templated insertions, which represent deletions carrying insertions that were of sufficient size (≥5bp) to be mapped to their origin in the immediate vicinity of the junctional sequence (<40bp surrounding the deletion breakpoints), iii) miscellaneous (misc.) insertions, which are deletions that also contain insertions, but which were below 5bp in size, or were of unknown origin. G) Heatmap representations of micro-homology present at deletion junctions, with a schematic representation illustrating the methodology: for every deletion, each position of the upstream breakpoint is compared to each position of the downstream breakpoint. Identical nucleotides score 1, non-identical score 0. Subsequently, a heat map is constructed by summing all scores for all events at each position divided by the number of events. For reference purposes, a heat map was constructed from all (368) deletions in the various genotypes, but with flank sequences shuffled *in silico*. Of note, all alleles are annotated in keeping with maximal 5' conservation, which here dictates that the base at the -1 position at the 5' side is never identical to the +1 position at the 3' side: in such a case, that base will shift to the +1 position at the 5' side. As a consequence of this rule, the position marked by a white square will have no microhomology score, while the +1,-1 position is slightly elevated. The extent of this methodological skewing can be noticed in the analysis of the random set of deletions. Number next to heatmap represents the fraction of micro-homology at the indicated location (NS—non-significant, ***—p<0.001, chi-square test compared to random set).

animals, a clear increase can be detected for deletions of size 50-500bp. This category of mutations, the product of TMEJ action on DNA breaks, are thus observed in all animals with compromised TLS, *i.e. polh-1*, *rev*-1 and *rev-3*, except in *polk-1* where POLH-1 can completely compensate for loss of POLK-1 (Fig 3D and Fig 3E). This class of deletions, with a narrow size range of 50-500bp, bears the typical signature of polymerase theta action upon DNA break repair: the occasional presence of templated insertions, and an overrepresentation of micro-homology at the deletion junction (Fig 3F and Fig 3G). In all these backgrounds also more complex events were found: a collection of inversions, tandem duplications or events that are more complex and classify into the umbrella term gross chromosomal rearrangements.

## Loss of TLS activity is compatible with *C. elegans* development and fertility

The observation that *rev-3* mutant animals have similar levels of deletion mutagenesis as *rev-1* mutant animals, together with the outcome that *rev-1* loss did not significantly elevate the genomic instability observed in *polh-1 polk-1* mutants predicts that *polh-1 polk-1 rev-1 rev-3* quadruple mutant animals can be generated, which are hence devoid of any TLS activity. Indeed, we found that the brood size and embryonic survival of these "TLS-dead" mutants, although lower than WT, is only mildly affected; animals also develop normally (Fig 3A and Fig 3B). We next determined the long-term genetic effects by monitoring mutational accumulation upon prolonged culturing (Fig 3D and Fig 3E, S1 Table, S2 Table). We found that *polh-1 polk-1 rev-1 rev-3* quadruplex mutant animals, similar to *rev-1* single mutant animals, displayed a reduced SNV mutation rate as compared to wild-type animals. However, the overall mutagenesis rate is greatly elevated, because of an approximately 30-fold increase in deletion mutagenesis, similar to the level observed in *polh-1 polk-1* animals (Fig 3D). This net increase in the mutational load in TLS-abolished conditions argues that bypass of endogenous lesions in TLS proficient animals is predominantly error-free. Of note, the observation that loss of REV-1 and REV-3 does not further increase the genomic instability observed in the *polh-1 polk-1*, argues that Polζ does not bypass endogenous replication fork barriers independent of Polη or Polκ (Fig 3D). Our data also suggests that the contribution of REV-1, as a scaffold, or REV-3, as an extender, is limited in the bypass of spontaneous and UV-induced DNA damage.

## Genome scarring in TLS deficient *C. elegans* is preferential at guanine bases

In an effort to deduce the molecular nature of the replication fork impediment from the mutation accumulation data we examined the base composition of deletion junctions. Previous work, analyzing deletions formed at replication-blocking G-quadruplexes, revealed that at least one of the breakpoints maps very close, if not immediately adjacent, to the replication impediment [30]. We reasoned that this may also occur at spontaneous lesions in TLS-compromised animals: one could argue that the nascent strand, which is extended by the replicative polymerases right up to the blocking lesion is a substrate for TMEJ when TLS cannot occur and thus defines one border of a deletion (Fig 4A). According to this logic, the first deleted base at the junction will frequently represent the nucleotide that is complementary to the damaged base. To assess this idea, we have determined the normalized base distribution at the deletion junctions for each genetic background. Because the total numbers of deletions in each single mutant was too low for statistically supported conclusions, we processed the data derived from *polh-1 polk-1*, *polh-1 polk-1 rev-1* and *polh-1 polk-1 rev-1 rev-3* mutant animals. In all three genetic backgrounds we detect a non-random distribution, *i.e.* a significant enrichment for cytosines at the -1 position (Fig 4B and S3B Fig; S3D Fig). This outcome is consistent with the idea that TLS is required predominantly at guanines to suppress replication-associated DSBs. Damaged guanines as the primary source for the deletion junctions also explains the apparent enrichment of cytosines at more downstream positions (-2 until -6) because TMEJ itself frequently results in loss of a few nucleotides at one or both sides of a DSB and micro-homology usage in TMEJ also perturbs correct annotation of the deletion junction (*e.g.* three different annotations are possible for a deletion with 2 bases of micro-homology) [34], which together dilutes underlying mechanistic parameters. We therefore also analyzed a subset of deletions for which such potential disturbing effects are reduced, *i.e.* deletions with insertions (see [34] for details). Using this subset, we indeed found the enrichment for cytosines at position -1 more pronounced (Fig 4C and S3C Fig; S3E Fig).

## Discussion

When the first eukaryotic factors of TLS (Rev1, Rev3 and Rev7) were characterized in yeast they were found to promote both induced and spontaneous mutagenesis. Here, we show that lack of TLS polymerases leads to an increase in the overall mutational load in *C. elegans* grown under unperturbed conditions, implying that TLS action suppresses mutagenesis. The major causes for the mutagenicity of TLS are the "hard-to-read" character of bulky lesions, the wider catalytic centers of TLS polymerases (allowing for less stringent base pairing), and the lack of proofreading. In bacteria and yeast, template switching provides an error-free alternative to TLS, which may explain the lower mutation rate in TLS deficient mutants [43,44]. While such a mechanism may be present in higher eukaryotes as well, its importance remains unclear [45]. Here, we have analyzed the contribution of TLS activity on long-term genetic stability in *C. elegans*. We monitored fitness and spontaneous mutagenesis in animals deficient for REV1, for REV3, for all Y-family TLS enzymes, and for all TLS activity. We found that *C. elegans* REV-1 and REV-3 both safeguard survival in *C. elegans* upon exposure to UV light but are not needed for development and reproduction under unchallenged conditions in a laboratory environment. To our surprise, we found that TLS is not essential for animal life as TLS-dead animals proliferate and produce morphologically normal progeny. While we found evidence for alternative processing of blocked replication, by means of TMEJ of DNA breaks, other biology may also contribute to the mild consequences of a complete TLS deficiency. For instance, when occurring in germline nuclei, DNA breaks can be repaired by homologous recombination, or DSB-containing nuclei can be removed through apoptosis. We found that Y-family-

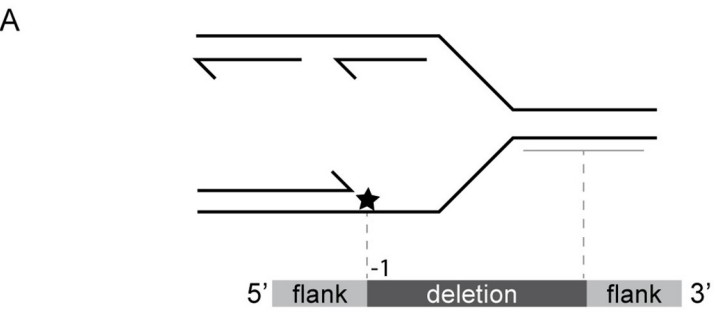

A

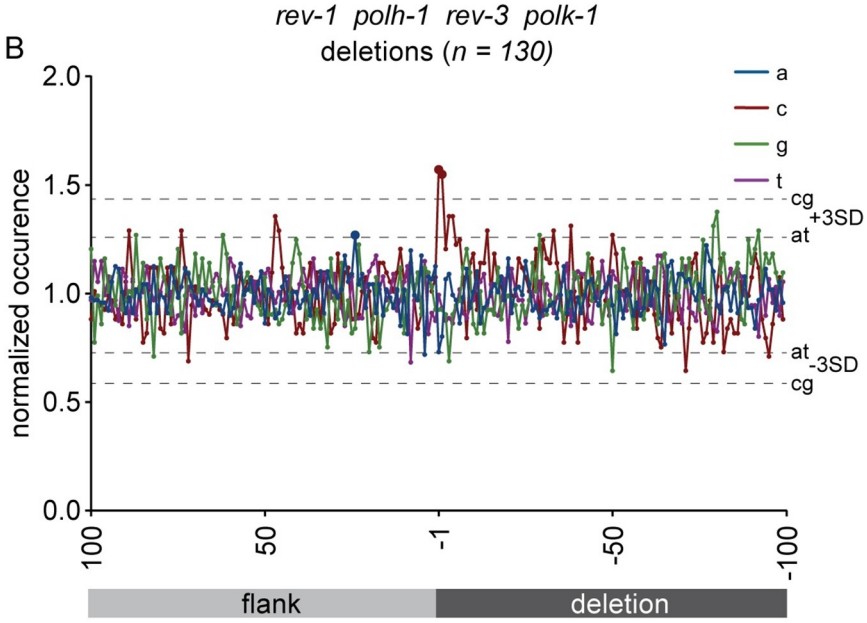

B

*rev-1 polh-1 rev-3 polk-1*
deletions (*n = 130)*

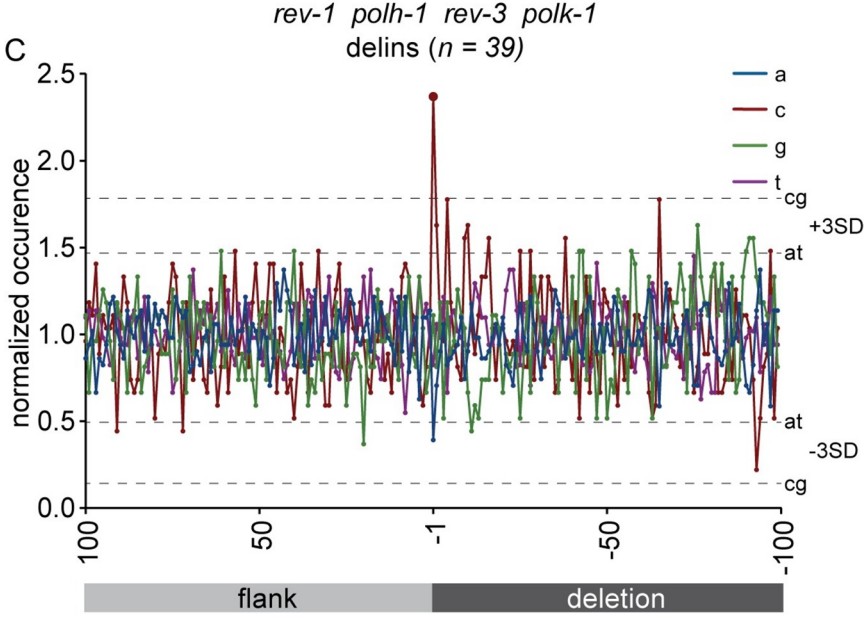

C

*rev-1 polh-1 rev-3 polk-1*
delins (*n = 39)*

**Fig 4. Failure to bypass damaged guanines results in deletions.** A) Schematic illustration of the concept that one junction of DNA-damage-induced deletions is defined by the nascent strand blocked at sites of DNA damage. In this hypothesis, the replication-blocking lesion may dictate position -1, being the outermost nucleotide of the lost sequence. B) The base composition of all deletion breakpoints, normalized to the relative AT/CG content around the breakpoints (from +100 to -100). Position 100 to 1 reflects the sequence that is retained in the deletion alleles; position -1 to -100 reflects the sequence that is lost. Dashed lines represent three times the SD. Data points outside these boundaries are marked with an enlarged dot. C) as in B, but now only for deletions with insertions.

dead and TLS-dead animals accumulate mutations over generations at the same rate and with the same characteristics as *polh-1 polk-1* mutant animals, while the increase in mutation accumulation in *rev-1* or *rev-3* animals is only minor, which argues that Polη and Polκ i) act redundantly to bypass the vast majority of spontaneous DNA damage in *C. elegans*, ii) act in an error-free manner, and iii) do not require the action of REV-1 nor REV-3 for most lesions. The comprehensive mutation profiles we present for different TLS compromised *C. elegans* strains support the idea that TLS is anti-clastogenic and we infer from our data that the most common endogenous replication blocks reside at guanines, which in the absence of TLS generate replication-associated DSBs that are repaired by polymerase theta-mediated end joining. This DSB repair pathway is intrinsically mutagenic, hence genome scars that are characterized by micro-homology are predicted to occur at sites where replication blocks persist. Of interest, such scars are also observed in tumors deficient for the breast cancer genes BRCA1 and BRCA2 (COSMIC mutational signature ID6). In those genetic contexts potential replication-dependent DSBs cannot be repaired in an error-free manner by homologous recombination and depend on alternative routes for their repair.

We found that deficiencies for REV-1 and REV-3 lead to very similar mutation profiles and UV-hypersensitivity, which supports the hypothesis that these proteins may act in concert. Their requirement during unchallenged *C. elegans* growth, however, is small: in their absence, ~0.1 deletion per generation is found, which, considering 10–15 cell divisions per generation argues that only 1 lesion per $10^{10}$ bases depends on REV-1 and REV-3 to be bypassed. The consequences of REV-1 or REV-3 loss in more complex animals is more detrimental. While REV1 deficient mouse embryonic fibroblasts, yeast and human fibroblast cell lines have no proliferative problems, Rev1[-/-] mice develop poorly [25, 45–48]. REV3 is essential for mammalian development: missense mutations in the human REV3L gene can cause Mobius syndrome which is associated with developmental abnormalities [49], and Rev3 knockout mice are embryonic lethal [50]. The fact that we do not observe such profound developmental and reproduction defect in worms may be explained by a low damage load and by the relatively small size of the *C. elegans* genome. Indeed, we found very low levels of spontaneous DSBs in *rev-1* deficient worms. Such replication-associated DSBs do not induce noticeable proliferative defects likely because they are, at least in part, repaired by TMEJ.

Another explanation for lack of viability in mammals when REV1 or REV3 function is lost may in part reside in functions outside TLS: in biology that may not be evolutionarily conserved. For instance, we cannot formally rule out a proposed role for Rev1 in homologous recombination but we have not found experimental support: defective HR in *C. elegans* results in reduced brood size, substantial or complete embryonic lethality and X-chromosomal non-disjunction [51], which we did not observe for *rev-1* mutants. Also, we have not found indications supporting a role for *C. elegans* REV-1 in replicating through G-quadruplex structures, as was observed in DT40 chicken B-lymphocyte cells [31,32]. *C. elegans* REV1 and REV3 proteins are substantially smaller than their mammalian counterparts and lack amino acids sequences, such as REV1's C-terminal domain, that manifest positive evolutionary selection pressure in vertebrates.

The frequency of deletion formation in the TLS-dead mutants may also provide an indication of how often a replication fork runs into an insurmountable block for the replicative polymerases in the context of a multicellular organism. In fact, if one assumes that in a TLS-dead mutant context every lesion that cannot be bypassed forms a deletion of >50bp, it becomes clear how rare spontaneous replication blocks are. In an estimated 10–15 rounds of replication per generation only 1 deletion occurs. For a genome of $10^8$ bases, this means that only 1 in ~$10^9$ bases requires TLS for its replication.

All our data point to damaged guanines as the predominant spontaneous lesion that requires TLS. Interestingly, OGG1, the glycosylase that in many other biological systems removes the majority of abundantly induced 8-oxo-G lesions via BER (reviewed in [52]), is not encoded by the *C. elegans* genome. This lack of repair together with this lesion being an efficient and accurate TLS substrate [53] may provide an explanation for this strong base effect. It is, however, not the case that the spontaneous mutations in *C. elegans* are predominantly G to A (or C to T) transitions, nor have we found a dramatic reduction of these types of mutations in TLS deficient worms. Together, our observations argue that the genome is *grosso modo* devoid of base damage when it is replicated, explaining the lack of TLS-induced mutations as well as the ability to reproduce when all TLS activity is lost. Despite the low damage load, we find that in absence of TLS the overall mutagenic consequence is strongly increased as small deletions (50 to 500 bp) manifest, which by virtue of being ORF disrupting have a more detrimental outcome than TLS–induced SNVs.

From our studies, we thus conclude that TLS in *C. elegans* not only protects replication potential, it also protects the genome against the formation of small genomic deletions and larger complex rearrangements and thereby helps to maintain genome stability on an evolutionary scale. These predominantly beneficial outcomes of TLS activity provides selective advantages for the presence of these specialized polymerases in all living organisms [8].

## Material & methods

### C. elegans genetics

All strains were cultured according to standard methods [54]. The N2 Bristol strain was used as WT control. The alleles *polq-1(tm2026), rev-1(gk455794), unc-91(e1500)* were obtained from the Caenorhabditis Genetics Center, Minnesota, USA. *The rev-1(BRCT)* allele was isolated via a random mutagenesis approach described in [55]. The *rev-1(lf206)* and *rev-1(fl207)* KO alleles were obtained via CRISPR/Cas9-mediated genome editing as described below in further detail.

### CRISPR/Cas9 mediated generation of *rev-1* null alleles

For CRISPR/Cas9 mediated targeting we used the following sequence (which includes the PAM site) in exon 2 of *rev-1*: AGTTTCATCCTCTTCGTCACTGG. Cloning was done as described in [56]. Plasmids were injected using standard *C. elegans* microinjection procedures. In brief, 1 day before injection, L4 animals were transferred to new plates and cultured at 15 degrees. Gonads of young adults were injected with a solution containing 20 ng/μl pDD162 (Peft-3::Cas9, Addgene 47549), 20 ng/μl pMB70 (u6::sgRNA with *rev-1* target, 10 ng/μl pGH8, 2.5 ng/μl pCFJ90 and 5 ng/μl pCFJ104. Progeny (F1) animals that express mCherry were picked to new plates 3–4 days post injection and allowed to produce offspring. Of each F1 plate 10 F2 animals were pooled, lysed and genotyped. Genotyping was done by PCR amplification of a 480 bp product around the CRISPR/Cas9 target site. Subsequent restriction with MaeIII enzyme of the wild-type sequence would result in 2 fragments (91 bp + 389 bp). A

mutation at the target site can disrupt the MaeIII recognition site resulting in an uncut PCR product. We isolated 4 alleles, 2 of which were small out-of-frame deletions (S1 Fig).

## Brood size and embryonic lethality assay

To determine the brood size, we singled L4 animals on OP50 plates. Every day for four days, we transferred the mother to a fresh plate and one day later quantified the number of embryos and larvae on the plate. We quantified the broods and embryonic lethality of > 20 animals per genotype.

## Survival assays

To measure germline sensitivity to UV, staged young adults (one day post L4) were transferred to empty NGM plates and exposed to different doses of UV-C. Per dose and genotype 3 plates with 3 adults were set up using NGM plates with OP50 and allowed to lay eggs for 24 hours. Subsequently adults were discarded and the brood on the plate was allowed to hatch. 24 hours later the number of non-hatched eggs and surviving progeny was determined.

## Sample preparation, RAD-51 antibody staining, imaging and quantification

Animals were synchronized by picking L4 stage worms 22h before dissection. Worms were dissected in 1x EBT (25 mM HEPES-Cl pH 7.4, 118 mM NaCl, 48 mM KCl, 2 mM $CaCl_2$, 2 mM $MgCl_2$, 0.1% Tween 20 and 20 mM sodium azide) to expose gonads. Most of the buffer was removed and the sample with cover glass was transferred to a Superfrost Plus slide and flash frozen on a metal block in dry ice. Upon complete crystallization of the sample the cover glass was quickly removed (freeze-cracking) followed by post-fixation in 4% PFA in 1x EBT. After washing, the samples were incubated with RAD-51 antibody (rabbit polyclonal from SDIX/Novus Biologicals, cat# 29480002, used at 1:1000 in PBST+0.5%BSA) overnight at room temperature. Alexa anti-rabbit 488, 1:500, was used as secondary antibody and incubated for 2h at room temperature. DNA was stained 10 min in 0.5 µg/ml DAPI/PBST. Finally, samples were de-stained in PBST for 1 h and mounted with Vectashield. Imaging and processing was done on a Leica DM6000 microscope. The data obtained (Fig 1D and Fig 1E) are from at least 3 independent experiments. Each data point represents an average value of the mitotic zone of one gonad.

## Mutation accumulation lines and whole genome sequencing

Mutation accumulation lines were generated by cloning out F1 animals from one homozygous hermaphrodite, generating subpopulations. Each generation three worms from each subpopulation were transferred to new plates. MA lines were maintained for on average 45 generations. Single animals were then cloned out and propagated to obtain full plates for DNA isolation. Worms were washed off with M9 and incubated for 2 hours while shaking to remove bacteria from the intestines. Genomic DNA was isolated using a Blood and Tissue Culture Kit (Qiagen). DNA was sequenced on an Illumina HiSeq machine according to manufacturer's protocol. Image analysis, base calling and error calibration were performed using standard Illumina software. Raw reads were mapped to the C. elegans reference genome (Wormbase release 235) by BWA [57]. GATK's HaplotypeCaller [58] was used for SNV calling. To identify larger indels and microsatellites, GATK [57] and Pindel [59] were used. In cases that only one program identified the structural variation, visual inspection was carried out using IGV [60]. Variations were marked as true if i) it is a *de novo* variation, thus all other subpopulations did not support

the variation. ii) covered by both forward and reverse reads, iii) covered at least five times. Mutation rates for different variations are calculated as the number of mutations per sample divided by the number of generations grown. See S1 Table for mutation rates and S2 Table for all identified *de novo* variations.

### G4 stability on qua1466

Qua1466 is a genomic sequence [GGGAGGGCGGGCGGG] located on chromosome IV (location 11,326,500–11,326,514) that can potentially form a G-quadruplex structure. To assay genomic instability and the formation of deletions at this site we performed a nested PCR reaction on lysed animals using the following primers: external forward CAAATAAGTATTGGG CCGAAACC; external reverse AAGGAACACCTTCAAGACTCC, internal forward CTGCG AACTTCTGACGAATTTG, internal reverse TTGACTCCTCCTCTTCTGGC. As template for the external PCR 1 µl of a 15 µl lysis with 5 worms was used. 0.5 µl of the external PCR was used as template for internal PCR. 10 µl of internal PCR product was run for 1 hour at 120V on a 1% agarose gel.

### *unc-93 (e1500)* mutagenesis assay

To pick up spontaneous mutations in the *rev-1(gk455794)* and *rev-1(gk455794) polq-1 (tm2026)* backgrounds, we used a mutagenesis assay based on reversion of the so-called "rubber band" phenotype, caused by a dominant mutation in the muscle gene *unc-93*. Reversion of the *unc-93(e1500)* phenotype is caused by homozygous loss of *unc-93(e1500)* or one of the suppressor genes *sup-9, sup-10, sup-11, and sup-18*. For both genotypes, *rev-1 unc-93(e1500)* and *rev-1; polq-1(tm2026); unc-93(e1500)*, 400 animals were singled on 9 cm plates. These plates were grown until starvation and of each plate an equal amount (chunks of 2 x 2 cm) were transferred to fresh 9 cm plates. Before these plates reached starvation they were inspected for wild-type moving animals. From each starting culture, only one revertant animal was isolated to ensure independent events. Of each genetic background we randomly selected 30 revertants and sequenced the *unc-93* gene. When large deletions occurred we established the approximate size of the deletion with PCR amplicons of approximately 500 bp located at the borders of the gene and 2.5 kb and 5 kb up and downstream of the *unc-93* gene.

## Supporting information

**S1 Fig. Schematic representations of all TLS polymerase genes encoded in *C. elegans* and the mutations used in this study.** A) *rev-1*, B) *polh-1*, C) *polk-1* and D) *rev-3*.
(PDF)

**S2 Fig. *rev-1* does not aggravate G-quadruplex deletions in *dog-1* deficient animals.** Nested PCR was performed on genomic G4 site qua1466 (expected size: 1000bp). 5 adult worms per well lysed in 15 ul lysis buffer. 1ul lysis per well was used as a template for a nested PCR. No additional genomic instability was observed in *dog-1* deficient animals after loss of REV-1. M denotes DNA marker. Wells were scored as positive when one or more bands appeared <1000bp. Empty wells were not included in the quantification.
(PDF)

**S3 Fig. Failure to bypass damaged guanines results in deletions.** A) Schematic illustration of the concept that one junction of DNA-damage-induced deletions is defined by the nascent strand blocked at sites of DNA damage. In this hypothesis, the replication-blocking lesion may dictate position -1, being the outermost nucleotide of the lost sequence. B-C) The base composition of all deletion breakpoints, normalized to the relative AT/CG content around the

breakpoints (from +100 to -100) in *polh-1 polk-1* (B) and *polh-1 polk-1 rev-1* (C). Position 100 to 1 reflects the sequence that is retained in the deletion alleles; position -1 to -100 reflects the sequence that is lost. Dashed lines represent three times the SD. Data points outside these boundaries are marked with an enlarged dot. D) as in B, but now only for deletions with insertions. E) as in C, but now only for deletions with insertions.
(PDF)

**S1 Table. Mutagenesis rates in *C. elegans* TLS mutants.**
(XLSX)

**S2 Table. Mutation profiles derived from *C. elegans* whole genome sequencing data.**
(XLSX)

## Acknowledgments

Some strains were provided by the CGC, which is funded by NIH Office of Research Infrastructure Programs (P40 OD010440).

## Author Contributions

**Conceptualization:** Marcel Tijsterman.

**Data curation:** Ivo van Bostelen, Robin van Schendel.

**Formal analysis:** Ivo van Bostelen, Robin van Schendel.

**Funding acquisition:** Marcel Tijsterman.

**Investigation:** Ivo van Bostelen, Ron Romeijn.

**Methodology:** Ivo van Bostelen, Marcel Tijsterman.

**Software:** Robin van Schendel.

**Supervision:** Marcel Tijsterman.

**Validation:** Ivo van Bostelen, Ron Romeijn.

**Visualization:** Ivo van Bostelen, Robin van Schendel.

**Writing – original draft:** Ivo van Bostelen, Marcel Tijsterman.

**Writing – review & editing:** Robin van Schendel, Marcel Tijsterman.

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
