## [Decision Letter · Decision Letter 0]

22 Mar 2020

Dear Dr Tijsterman,

Thank you very much for submitting your Research Article entitled 'Translesion synthesis polymerases are dispensable for C. elegans reproduction but suppress genome scarring at sites of base damage' to PLOS Genetics. Your manuscript was fully evaluated at the editorial level and by independent peer reviewers. The reviewers appreciated the attention to an important topic but identified some aspects of the manuscript that should be improved.

We therefore ask you to modify the manuscript according to the review recommendations before we can consider your manuscript for acceptance. Your revisions should address the specific points made by each reviewer but no new wet-lab experiments are necessary.

[LINK]

Yours sincerely,

Julie Ahringer

Associate Editor

PLOS Genetics

Gregory P. Copenhaver

Editor-in-Chief

PLOS Genetics

Reviewer's Responses to Questions

**Comments to the Authors:**

Reviewer #1: Bostelen et al. investigate the consequence of translesion synthesis polymerases in mutagenesis in C. elegans. Translesion synthesis (TLS) polymerases are required for replicating DNA amid damaged bases. Studies in yeast and bacteria have established that TLS polymerases are error prone. Bostelen et al. now have analyzed TLS polymerase mutants in C. elegans to better understand to which extend they are responsible for replication-associated mutagenesis in metazoan.

They start with the analysis of several alleles of rev-1 that confer UVC hypersensitivity but are dispensable for fitness under unperturbed laboratory conditions even though RAD-51 foci indicative of DSB formation were elevated in replicating germ cells. Single nucleotide variants (SNVs) were reduced while deletion number and size were strongly elevated in rev-1 mutants. Furthermore, embryonic lethality of polq-1 is exacerbated by a rev-1 mutation. However, the deletion frequency of polh-1;polk-1 double mutants was not altered by a rev-1 mutation and rev-1 and rev-3 were epistatic. Analysis of the deletion sequence revealed that particular C at -1 position were enriched indicating that deletions occurred amid damaged guanines.

Taken together, the authors provide a thorough mutation analysis of TLS polymerases that establish that they primarily protect the genome from deletions in C. elegans. Their study makes an important contribution to better understand the consequences of TLS particularly in the metazoan context and will be of interest to the community across the fields of DNA repair, replication, and genetics.

Specific comments:

1. It would be helpful if the authors could summarize their analyses of mutations per generation in a table that would also include literature data as comparison as some of the single TLS mutants are referred to in the literature and not analyzed again here.

2. SD should be used instead of SEM whenever distinct populations (e.g. mutant vs. wildtype) are compared.

Reviewer #2: The manuscript by van Bostelen and colleagues investigates the role of translesion synthesis in genome stability using C. elegans as a model system. Continuing from their previous results, the authors monitored the fitness, DNA damage resistance and spontaneous mutagenesis in strains mutant for individual TLS polymerase genes, or their combinations. According to their somewhat surprising findings, rev1 and rev3 knockouts only weakly influence spontaneous mutagenesis. The generation of SNVs is reduced in the rev1, but not in the rev3 mutant (in contrast with yeast), and there is an increase in the formation of >50bp deletions, but this increase is moderate compared to the simultaneous knockout of polh and polk.

The data shed new light on the relative roles of TLS polymerases in causing base substitution mutations, and in protecting from DNA breaks. The manuscript also contains a number of useful specific observations, for example that deletions (and the originating DNA breaks) may initiate precisely at the sites of damaged bases, as suggested by the sequence preference of the start of deletions. The analysis is of good quality, with little to add.

The main weakness of the approach is that in the small worm genomes very few mutations arise, even after many generations. For example, the SNV mutation spectra in fig.1B cannot be accurately compared due to the very low number of events, and the SNV numbers are also not sufficient to generate ‘triplet’ mutation spectra and compare them to cancer-derived mutation signatures. It would have been very useful had the authors chosen to sequence a larger number of animal lineages.

Further comments:

Introduction: When listing the TLS polymerases, please also make a mention of the TLS role of pol delta, especially PolD3 (see e.g. PMID: 25628356).

p6 Fig. 1F unc-93 reversion assay: again, it would be great to have some more sequences. The authors should also analyse the context of the deletions, specifying if they show microhomology.

p9: The polh polk rev1 triple mutant had marginally fewer mutations than the polh polk double mutant, indicating a weak role for rev1 in this background. But considering that a major function of rev1 is the recruitment of other Y family polymerases rather than DNA synthesis, this could also be an epistatic relationship. The recruitment role of rev1 is discussed in the next section, but only with respect to pol zeta. It is possible that rev1 is needed for both polh/polk recruitment and polz recruitment, and both of these are needed for lesion bypass according to the two-polymerase model.

Alternatively, the explanation for the much stronger phenotype of the polh polk double mutant compared to rev1 or rev1 rev3 could suggest that polh and polk are primarily recruited through a rev1-independent mechanism, presumably PCNA ubiquitylation. This mechanism has been investigated recently (PMID: 31450086) and should be discussed.

In the discussion, please further emphasise that the results are C. elegans specific, and compare and contrast them with yeast and vertebrate data where available. Please compare the spectrum of deletions in the TLS mutants to the recently published COSMIC version 3 indel signatures.

Importantly, there needs to be a supplementary table for the WGS data, showing the precise number of worms/worm populations sequenced, the number of generations, the number and classification of mutations in each sample, and the list of mutations (with chromosomal position) for each sample. I cannot find any of this information in the manuscript. It is not sufficient to share the raw sequence files.

It would also be useful to show a summary of the mutation data in a simple table in the text. Currently these data are split to numerous graphs, with different scales, making comparisons difficult.

These tables should also include previously published data that was re-analysed here.

The doi link of ref 17 is incorrect.

Reviewer #3: Review of von Bostelen et al 2020

In this work, the authors build on their prior work analyzing viability, UV sensitivity and mutation frequency and spectrum in C. elegans mutants lacking translesion synthesis (TLS) polymerase activities and/or proteins. In previous work , they had analyzed a subset of single and double mutants lacking activity of TLS polymerases. Here, they add analysis of mutants lacking Y-family polymerase REV1 and/or the REV3 catalytic subunit of the B-family TLS polymerase, as well as mutants deficient for activities of: 1) all Y family polymerases, or 2) all TLS polymerases. They show that most animals lacking Y or all TLS polymerases are viable, but accumulate spontaneous deletions (with or without associated insertions) with size ranges distinct from those seen in wild-type worms and with signatures that are suggestive of repair of DSBs via a microhomology-mediated end-joining pathway (TMHEJ). Further, they provide indirect evidence, based on an excess of C residues adjacent to the likely site of DNA breakage inferred from deletion breakpoint analyses, suggesting that guanine adducts are the predominant types of lesions whose repair depends on TLS polymerases. In addition, an observed increase in mutation frequency in TLS-null or Y-polymerase null mutants suggests that repair mediated by TLS polymerases in the C. elegans germ line is substantially error free.

Overall, the experiments and analyses presented here are largely sound and well documented, and the topic is certainly appropriate for PLoS Genetics. However, the work is substantially weakened by statements made throughout the manuscript that conflate actual experimental findings with interpretations. That is, the authors claim to have demonstrated Y, when in fact they have actually demonstrated X, and Y is the favored interpretation. This failure to make rigorous distinctions between: a) experimental results and b) interpretations and conclusions based on the results is problematic for several reasons. It is misleading to more naïve readers who may not recognize the conflation, and it is also misleading to busy expert readers who may assume that a different type of assay must have been performed in order to reach the stated conclusion. For example, the authors have a tendency to make statements that would appear to require biochemistry to support them. The authors must be careful to articulate their conclusions in a manner that aligns with the actual assays performed and the type of data from which they are derived.

The above-described problem, while pervasive and serious, can be addressed almost entirely by textual revisions. Many instances are flagged here, but please note that this listing is not comprehensive, and the authors should make appropriate revisions throughout the manuscript.

Title:

“suppress genome scarring at sites of base damage”

It is true that TLS polymerases (collectively) suppress formation of spontaneous deletions. However, it has not been demonstrated that the inferred breaks that give rise to the deletions occur at sites of base damage.

Abstract:

It is shown that the TLS polymerases antagonize the formation of deletions, but the idea that they prevent DSBs from occurring is an indirect inference. And again, the idea that breaks are occurring at sites of base damage is hypothesized based on biochemistry in other systems, but it not demonstrated here; nor is directly demonstrated that guanine adducts are the main source of DNA damage requiring TLS for bypass (see comments below regarding discussion), although the data are consistent with this interpretation. Statements made in the abstract must make a clear distinction between what is shown, and what are (likely) interpretations supported by the data. It is entirely appropriate to interpret data in light of a model, as long as the distinctions are clear.

The last sentence seems like an odd closer given the statement just prior that TLS in C. elegans is predominantly error free- can the authors articulate: where is the risk that needs to be outweighed?

Results:

P5 l 102-109: 1) cannot equate RAD-51 foci with DSBs; RAD-51 foci do NOT necessarily represent DSBs, as RAD-51 can also associate with other types of lesions, e.g. ss gaps.

2) “apparently, these spontaneous DSBs do not affect proliferation…” Based on the info provided, one cannot assume repair- if lesions such as breaks or extensive ss DNA persisted into meiotic prophase, they would almost certainly be detected by DNA damage checkpoints and eliminated by apoptosis. Note that the authors could potentially test whether apoptosis is elevated in these mutants, but given that it is estimated that >50 of oogenic germ cell nuclei are eliminated during WT meiosis, even if there were a significant increase in nuclei harboring unrepaired DNA lesions, this might not be sufficient to move the needle on apoptosis frequency. The best thing to do here would be to simply remove the problematic narrative and jump into the compelling evidence for altered mutation profiles.

P6, Fig 1:

l 108, state “asked whether mutation induction is elevated”- but while the assay in this section evaluates mutation profile, insufficient information about mutation frequency is provided here. Please elaborate on effects on mutation frequency, or if there are concerns that the unc-93 mutation assay was performed in a way that would not provide reliable information regarding frequency, modify the set-up and other text to focus this section on mutation profile.

Statistically analysis (e.g. Fisher exact test) evaluating the relative numbers of SNPs vs indels among the unc-93 mutations from WT vs revBRCT mutant indicate that there is not statistical support to conclude that the mutation profiles are different.

“From this data we conclude…..bypass of endogenous lesions”: This is NOT a lesion bypass assay, please align conclusion with the assay.

“… role in SUPRESSING spontaneous mutagenesis LIKELY by suppressing the formation…”

P7 Fig 2

Please provide more information (could be in Figure legend or methods) regarding how mutation rates were deduced from the mutation accumulation lines.

L 145 “…identified 15 cases…” Here, the point scan actually be made more strongly- rather than calling out specifically one deletion size class, the authors can point out that the whole size distribution is very different (2C), a result with very strong statistical support with Mann-Whitney test (p value should be provided). Suggest stating: “There is both an increase in the incidence of deletions and a substantial difference in the size distributions of the deletions observed.”

L148 To be able to make a statement about a class of mutations being present in one genotype but absent in another, need to provide information about the numbers and to address whether there is statistical support for there being a real difference between genotypes.

P8 l 175-178

Here the set-up narrative is flawed- one can conclude that repair of UV-induced lesions in the absence of REV1 is Pol theta dependent, CONSISTENT WITH…(interpretation).

P9 L194. Here, and elsewhere (and possibly in the Discussion), the authors need to consider the possibility that elimination of damage-containing cells via apoptosis could profoundly affect the embryonic survival metric; at a minimum, they need to discuss this issue as it is relevant to the conclusion that TLS polymerases are non-essential.

L199-202. Avoid making statements when there is not statistical support.

Fig 2F-H, Fig 3 (all except G and maybe F)

To facilitate comparisons among these genotypes, the data in these panels should be consolidated so that there is one graph of each type:

Brood size

Survival

UV sensitivity

Mutations/generation

Deletion sizes

(Possibly Indel type pie charts, although those could be deferred to the next figure.)

(It is ok if not all genotypes are represented in every graph type)

The narrative in the Results section can still consider subsets of the data sequentially as the picture is being built up, as in the current version.

Fig 3G

3G and 3 F could potentially stand alone as a figure by themselves (new Fig 4), or they could be the last 2 panels of the new revised Fig 3,

However it is done, that the heat map for the random in silico generated deletions should be the same size as the real data for better comparison. Also, the actual values in the +1 -1 position should be indicated below the graph for each genotype and random. Finally, some evidence regarding whether there is statistical support for microhomology should be provided. One simple way would be to use a Chi-square test to assess, for the +1 -1 position, whether the incidences of same vs different nucleotide in the real deletions observed for each genotype differ from those expected for random. (Based on the visual representation and low numbers for rev3 single mutant, there may not be statistical support for this genotype; if that turns out to be the case, the text should be revised appropriately.)

P 10 L234-5. Change order of sentence to avoid semantic ambiguity: “”…repair, micro-homology at the deletion junction (G) and the occasional presence of…(F)”

P 11 L253-5. This sentence is too definitive- need alternative wording that doesn’t overstate. E.g. need to consider/discuss the possibility that more lesions might form under these mutant conditions, and/or if this is considered unlikely, explain why.

P 11 L261. Section heading is too definitive- even though it is a reasonable explanation for the result, the section heading should state the finding, not the interpretation.

Discussion:

P 12 The discussion opens with the statement that TLS polymerases in yeast were found to “promote both induced and spontaneous mutagenesis”

HERE, the authors show that LACK of TLS polymerases apparently increases mutagenesis, implying that the TLS polymerases (as least as a cohort) INHIBIT mutagenesis. This apparent contrast should be addressed explicitly and head-on.

P13 L308 -314. This is a model, with features that are consistent with the data. The authors need to more clearly distinguish between what is shown and models that are supported by the findings.

The data clearly show that the TLS polymerases are anti-clastogenic. They also show that Cs are overrepresented adjacent to deletion junctions. These data are consistent with a model in which: 1) breaks that give rise to the observed deletions tend to occur at or near the sites of damaged bases that cause replication blockage and 2) damaged Gs are the most common replication blockages that require TLS polymerases for their bypass.

The discussion of amounts of spontaneous DSBs on P 14 completely ignores the possibility of elimination of cells with DSBs by apoptosis and/or use of HR to repair such breaks. These are relevant to the discussion, as both HR and DNA damage checkpoints are highly active in the germline.

I’m a bit confused by the last sentence of the discussion- hasn’t it already been argued that bypass using the TLSs in C. elegans is predominantly error free ( based on an increase in overall mutation frequency when they are absent)? If so, where is the trade-off?

**Have all data underlying the figures and results presented in the manuscript been provided?**

Reviewer #1: Yes

Reviewer #2: No: The raw data referenced in the paper is not yet available in the SRA. I made suggestions above for including processed mutation data in the supplementary material.

Reviewer #3: Yes

PLOS authors have the option to publish the peer review history of their article (what does this mean?). If published, this will include your full peer review and any attached files.

Reviewer #1: Yes: Björn Schumacher

Reviewer #2: No

Reviewer #3: No

---

## [Decision Letter · Decision Letter 1]

6 Apr 2020

Dear Dr Tijsterman,

We are pleased to inform you that your manuscript entitled "Translesion synthesis polymerases are dispensable for C. elegans reproduction but suppress genome scarring by polymerase theta-mediated end joining" has been editorially accepted for publication in PLOS Genetics. Congratulations!

Please note that Reviewers #2 and #3 had a couple of helpful suggestions (see below) that you might want to consider as you prepare your final draft for the production team (the editorial team will not need to re-evaluate).

Yours sincerely,

Julie Ahringer

Associate Editor

PLOS Genetics

Gregory P. Copenhaver

Editor-in-Chief

PLOS Genetics

Comments from the reviewers (if applicable):

Reviewer's Responses to Questions

**Comments to the Authors:**

Reviewer #1: The authors adequately addressed all my comments.

Reviewer #2: I thank the authors for the changes and have no further scientific comments.

Minor issues:

I suggest adding a colour key to the new Figure 3D panel.

Fred Sanger did not write his name with an umlaut (Fig. 2 legend).

Reviewer #3: This revision very nicely addresses the issues raised in the previous review. I especially appreciate how much easier it was to digest and compare the information for different mutant genotypes in the newly consolidated Figure 3. I am happy to recommend acceptance of this revised manuscript.

In preparing the final files for publication, please address the following:

1) Add color key to Fig 3D graph

2) Several remaining typos (e.g. “Off note”, on line 245) or small grammatical errors in the manuscript text. Since PLoS Genetics does not use a copy editor, it will be helpful to have a fresh pair of eyes to catch these.

**Have all data underlying the figures and results presented in the manuscript been provided?**

Reviewer #1: Yes

Reviewer #2: Yes

Reviewer #3: Yes

PLOS authors have the option to publish the peer review history of their article (what does this mean?). If published, this will include your full peer review and any attached files.

Reviewer #1: Yes: Björn Schumacher

Reviewer #2: No

Reviewer #3: No

**Data Deposition**

http://datadryad.org/submit?journalID=pgenetics&manu=PGENETICS-D-20-00260R1

**Press Queries**

---

## [Editor Report · Acceptance letter]

17 Apr 2020

PGENETICS-D-20-00260R1 

Translesion synthesis polymerases are dispensable for C. elegans reproduction but suppress genome scarring by polymerase theta-mediated end joining 

Dear Dr Tijsterman, 

We are pleased to inform you that your manuscript entitled "Translesion synthesis polymerases are dispensable for C. elegans reproduction but suppress genome scarring by polymerase theta-mediated end joining" has been formally accepted for publication in PLOS Genetics! Your manuscript is now with our production department and you will be notified of the publication date in due course.

With kind regards,

Jason Norris

PLOS Genetics

On behalf of:
